# Hypoxia Induces DPSC Differentiation versus a Neurogenic Phenotype by the Paracrine Mechanism

**DOI:** 10.3390/biomedicines10051056

**Published:** 2022-05-03

**Authors:** Simona Delle Monache, Fanny Pulcini, Francesca Santilli, Stefano Martellucci, Costantino Santacroce, Jessica Fabrizi, Adriano Angelucci, Maurizio Sorice, Vincenzo Mattei

**Affiliations:** 1Department of Biotechnological and Applied Clinical Sciences, University of L’Aquila, 67100 L’Aquila, Italy; fanny.pulcini@graduate.univaq.it (F.P.); adriano.angelucci@univaq.it (A.A.); 2Biomedicine and Advanced Technologies Rieti Center, “Sabina Universitas”, 02100 Rieti, Italy; f.santilli@sabinauniversitas.it (F.S.); s.martellucci@sabinauniversitas.it (S.M.); c.santacroce@sabinauniversitas.it (C.S.); j.fabrizi@sabinauniversitas.it (J.F.); 3Department of Experimental Medicine, “Sapienza” University of Rome, 00161 Rome, Italy; maurizio.sorice@uniroma1.it

**Keywords:** DPSCs, neuronal differentiation, hypoxia, stem cells, dental pulp

## Abstract

As previously described by several authors, dental pulp stem cells (DPSCs), when adequately stimulated, may acquire a neuronal-like phenotype acting as a favorable source of stem cells in the generation of nerves. Besides, it is known that hypoxia conditioning is capable of stimulating cell differentiation as well as survival and self-renewal, and that multiple growth factors, including Epidermal Growth factor (EGF) and basic fibroblast growth factor (bFGF), are often involved in the induction of the neuronal differentiation of progenitor cells. In this work, we investigated the role of hypoxia in the commitment of DPSCs into a neuronal phenotype. These cells were conditioned with hypoxia (O_2_ 1%) for 5 and 16 days; subsequently, we analyzed the proliferation rate and morphology, and tested the cells for neural and stem markers. Moreover, we verified the possible autocrine/paracrine role of DPSCs in the induction of neural differentiation by comparing the secretome profile of the hypoxic and normoxic conditioned media (CM). Our results showed that the hypoxia-mediated DPSC differentiation was time dependent. Moreover, conditioned media (CM derived from DPSCs stimulated by hypoxia were able, in turn, to induce the neural differentiation of SH-SY5Y neuroblastoma cells and undifferentiated DPSCs. In conclusion, under the herein-mentioned conditions, hypoxia seems to favor the differentiation of DPSCs into neuron-like cells. In this way, we confirm the potential clinical utility of differentiated neuronal DPSCs, and we also suggest the even greater potential of CM-derived-hypoxic DPSCs that could more readily be used in regenerative therapies.

## 1. Introduction

Dental pulp is a non-mineralized oral tissue composed of soft connective tissue, vascular, lymphatic components, and nerve elements located in the central pulp cavity of each tooth, and it contains different types of cells [1,2,3]. Recent studies have shown that dental pulp is considered a new accessible and rich source of stem cells, with mesenchymal stem cell (MSC) characteristics. MSCs from dental pulp (DPSCs) were isolated for the first time within the “cell rich zone” of the third molar dental pulp tissue [3]. They are ectoderm-derived stem cells, originating from migrating neural crest cells, and they represent a heterogeneous population with high plasticity and multipotency [4]. Their properties are linked to the dental pulp tissue origin that occurs from all three embryonic layers: the ectoderm, mesoderm, and endoderm [5]. Most importantly, DPSCs possess the same in vitro properties as MSCs, and present high proliferative, self-renewal, and multi-lineage differentiation potential [6,7,8,9,10]. In fact, in response to local stimuli, DPSCs can differentiate in several cell populations such as odontoblasts, osteoblasts, neural cells, chondrocytes, adipocytes, myoblasts, fibroblasts, endothelial cells and pericytes [11,12,13,14,15,16]. For this reason, differentiated DPSCs have the potential to be used in tissue regeneration [17,18]. Initially, DPSCs were used for the regeneration of dental tissue only, demonstrating the ability of DPSCs to differentiate into odontoblasts. Later, it was found that DPSCs are not only able to mediate reparative dentinogenesis, but also to promote both the in vitro and in vivo repair of various dental and periodontal tissues [19,20,21,22,23]. Nowadays, DPSCs represent a promising source of MSCs for tissue repair, even outside the tooth, and so other differentiation pathways have been investigated in recent years. Several studies have reported DPSCs’ ability to differentiate into adipogenic cell lineages under specific culture conditions, displaying changes in differentiation status both by Oil-Red-O staining, which highlights the accumulation of lipid droplets in cultures, and by analyzing the expression of early and late specific adipocyte genes such as *PPAR-A2* and *ap2* [24,25,26,27,28,29]. Furthermore, DPSCs show similar surface markers and matrix proteins associated with the formation of mineralized tissue (alkaline phosphates, osteocalcin, and osteopontin), and they represent a possible resource candidate to be differentiated in osteocytes/osteoblasts [30,31]. Applying specific culture conditions, DPSCs could find further applications in chondrogenic lineage differentiation. Chondrogenic differentiation, under specific culture conditions, is another important lineage in which DPSCs could find different applications [32]. In 2012, Dai et al. co-cultured human costal chondrocytes (CCs) with DPSCs, and their results showed that the CCs were able to supply a chondro-inductive niche that induced the DPSCs to undergo chondrogenic differentiation and enhance the formation of cartilage [33]. Moreover, recent studies have shown that DPSCs can also differentiate into neuron-like cells [3,34]. DPSCs are known to express nestin, a protein detectable in dividing cells during early development in the central nervous system (CNS), as well as peripheral (SNP), myogenic, and other tissues, and low basal levels of markers associated with mature CNS cell types, including the neuronal markers microtubule-associated protein 2 (MAP2), neurofilament (NF), β3-tubulin, CNPase associated with oligodendrocytes, and astrocytic marker glial fibrillary acid protein (GFAP) [35,36]. This pattern suggests that DPSC populations can indeed differentiate into neuron-like cells under appropriate conditions exhibiting typical electrophysiological properties [14,37,38]. In recent years, DPSC neuronal differentiation has been demonstrated in several in vitro studies using different approaches and methods. Various studies have shown that the bFGF can enhance peripheral nerve regeneration after injury, and phospho-ERK (p-ERK) activation as a major mediator may be involved in this process [39,40]. Rafiee et al. evaluated DPSC neuronal differentiation using stimuli with both epidermal growth factor (EGF) and bFGF [41]. Additionally, an interesting trophic factor is the nerve growth factor (NGF), which has been reported to improve cholinergic function, stimulate axonal growth, cerebral perfusion, and neurogenesis by stimulating proliferation through tyrosine kinase receptor signaling [42,43]. Another important factor that should be taken into consideration during MSC differentiation is the oxygen (O_2_) concentration in the culture environment, which seems to play an essential role in maintaining stem cell plasticity and proliferation [44]. In fact, human cells can sense and respond to O_2_ fluctuations as they change from normal (normoxia) to subnormal levels (hypoxia) [45]. Hypoxia is defined as a state of low availability of O_2_ that limits or even abolishes the functions of organs, tissues, and cells [46]. Although hypoxia is commonly studied related to a wide variety of pathological conditions such as tissue ischemia, inflammation, and tumors, it is known to play an important role in physiological processes such as embryogenesis and the regulation of stem cell proliferation and differentiation [46,47]. In recent studies, researchers have demonstrated the effects of hypoxia on DPSCs’ stemness [48,49]. The understanding of how O_2_ levels modulate the cellular pathways involved in DPSC neuronal differentiation could be a crucial step towards developing novel strategies for cell therapy and for neurogenesis. Furthermore, as recent studies suggest that the therapeutic activity of MSCs is due to the release of their paracrine factors in CM, their use can be supposed to a wide range of regenerative therapies such as myocardial infarction, stroke, bone regeneration, hair growth, and wound healing [50,51]. CM from differentiated neuronal DPSCs can be inoculated in vivo with therapeutic effects in neurodegenerative disease models, as recently performed with exfoliated deciduous stem cells (SHEDs) for the treatment of retinal degeneration [52]. Therefore, the use of DPSC-derived CM (DPSC-CM) could support the increase in the paracrine factor gradient between the diseased organ and the stem cell niche accelerating the healing process [53,54,55]. So, the aim of our study was to examine and evaluate the effects that hypoxia (O_2_ 1%) may directly exert on DPSCs, and the role of DPSC-CM obtained after hypoxia conditioning at different intervals on the DPSC neuronal differentiation process in vitro.

## 2. Materials and Methods

### 2.1. Cell Culture

#### 2.1.1. DPSCs

DPSCs) were purchased from Lonza (Walkersville, MD, USA) and cultured in Dental Pulp Stem Cell BulletKit™ Medium, which includes both basal medium and the necessary supplements for human dental pulp mesenchymal stem cell proliferation (Lonza, Walkersville, MD, USA), in a humidified incubator under an atmosphere of 5% CO_2_ at 37 °C. The culture medium was replaced every 3 days and when 90% confluence was achieved, cells were harvested using 0.05% Trypsin-EDTA (Euroclone, Milan, Italy). Cells were cultured with between 4 and 8 passages for the subsequent experiments, and each was repeated at least three times.

#### 2.1.2. SH-SY5Y

*Human* neuroblastoma cell line (SH-SY5Y) was cultured as a monolayer in polystyrene dishes and maintained in DMEM with high glucose formula (Gibco BRL) containing 10% of fetal bovine serum (FBS) (Euroclone, Milan, Italy), 100 IU/mL penicillin, and 100 µg/mL streptomycin (Euroclone, Milan, Italy) in a humidified incubator under the atmosphere of 5% CO_2_ at 37 °C. When 85% confluence was achieved, cells were harvested using 0.05% Trypsin-EDTA (Euroclone, Milan, Italy) and were passaged. The culture medium was replaced twice a week.

### 2.2. DPSC Proliferation Assay

A proliferation assay was performed to evaluate the effect of hypoxia on DPSC growth. Stem cells were seeded in a 96-well plate (Euroclone, Milan, Italy) at a density of 5 × 10^3^ cells/well and kept in a hypoxia incubator at continuous low oxygen tension (1%) for 5 and 16 days. Cell proliferation was evaluated by staining the cells with crystal violet (1%) (Merk Life Science, Sigma Aldrich, Milan, Italy ) and solubilized using a solubilization solution (1% SDS and 50% methanol). The plate was read at 595 nm.

### 2.3. Phalloidin Staining of the DPSCs

To visualize the effect of hypoxia on the cellular cytoskeleton, DPSCs were stained with Phalloidin. Briefly, cells were seeded in 6-well plates at 2 × 10^5^/well density, and grown for 24 h. Afterwards, cells were kept in a hypoxia incubator at continuous low oxygen tension (1%) or maintained in normoxia for 5 and 16 days. At the end of the experiment, the cells were fixed with 4% paraformaldehyde (PFA) (Euroclone, Milan, Italy), for 10 min. Non-specific binding sites were blocked with 3% bovine serum albumin (BSA) (Euroclone, Milan, Italy) and 0.1% Triton X-100 (Merk Life Science, Sigma Aldrich, Milan, Italy) for 30 min, and then DPSCs were labeled with FITC-Phalloidin (Fluorescein Isothiocyanate) (Merk Life Science, Sigma Aldrich, Milan, Italy) 1:250 in phosphate-buffered saline (PBS) for 1 h at room temperature, and then washed three times with PBS. Finally, nuclei were counterstained with Hoechst solution (Euroclone, Milan, Italy) and the images were acquired with a Zeiss Axio Vert.A1 fluorescence microscope (Zeiss, Milan, Italy).

### 2.4. DPSC Hypoxic Culture Regime

To evaluate the effect of hypoxia on the DPSC neuronal differentiation process, DPSCs incubated under O_2_ 1% condition were compared to DPSCs maintained in normoxia at the same times interval as indicated in Table 1. In brief, DPSCs were cultured in continuous low oxygen tension (1%) in a hypoxia incubator for 5 and 16 days. For normoxic conditions, the DPSCs were grown in a standard incubator under O_2_ atmosphere conditions and 5% CO_2_ at 37 °C for the same times. All experimental groups were analyzed for morphology, proliferation, and neurite outgrowth ability.

### 2.5. In Vitro DPSC Neuronal Differentiation

To induce neuronal differentiation, DPSCs were cultured in appropriate induction media, as previously described [56,57]. In this case, the cells were cultured and used between 4 and 8 passages, and subsequently stimulated with Neurobasal A medium containing L-Glutamine, supplemented with B27 (Life Technologies, Monza, Italy), bFGF 40 ng/mL, and EGF 20 ng/mL (PeproThec, DBA, Milan, Italy) for 14 days. The induction media was replaced every 3 days [14].

### 2.6. Preparation of the Conditioned Media from Hypoxic Culture

To evaluate the role of DPSCs’ secretome in the neuronal differentiation process, the culture medium of DPSCs at 4–5 passages was changed, and cells were resuspended in serum-free media for 48 h in a humidified incubator under the atmosphere of 5% CO_2_ at 37 °C, prior to experimentation. The conditioned medium (CM) from each sample in different treatment conditions (Table 1) was then centrifuged at 2600× *g* at 4 °C for 10 min, and then the collected CM, normalized with respect to cell number, was either used immediately or frozen at −80 °C until needed for the experiments. The DPSCs’ CM from the cells cultured in treatment conditions for 5 and 16 days was used to stimulate SH-SY5Y and DPSCs for 48 h and 10 days, respectively, as reported in Table 2.

### 2.7. Flow Cytometry Analysis

DPSCs in different treatment conditions (Table 1), SH-SY5Y stimulated with DPSCs’ CM and DPSCs stimulated with their own CM (Table 2) were analyzed by flow cytometry. For each experiment, at least 1 × 10^5^ cells were fixed with 4% paraformaldehyde for 10 min at 4 °C and permeabilized with permeabilizing solution (0.1% Triton X-100 in PBS for 10 min at 4 °C. After washing, cells were subsequently incubated over night at 4 °C with various primary antibodies, including: mouse Anti-CD44 mAb; mouse Anti-CD90 mAb (included in a Human MSC Analysis kit; Merck, Millipore, Milan, Italy); mouse Anti-CD105 mAb (BD Biosciences, Milan, Italy); mouse Anti-STRO1 mAb and mouse Anti-CD73 mAb (Merck, Millipore, Milan, Italy); mouse Anti-nestin mAb; mouse Anti-B3-Tubulin mAb; mouse Anti-NFH mAb; and rabbit Anti-GAP43 mAb (Cell Signaling Technology, Danvers, MA, USA). Next, cells were washed, centrifuged, suspended in 50 μL of PBS (Euroclone, Milan, Italy), and stained by PE-conjugated Anti-mouse IgG H&L and Anti-Rabbit IgG H&L (Cy5) (Abcam, Cambridge, MA, USA) in the dark for 30 min. All samples (at least 20,000 events were acquired) were analyzed with a FACScan cytometer (BD Accuri C6 Flow cytometer) equipped with a blue laser (488 nm) and a red laser (640 nm) (BD Biosciences, Milan, Italy). The same analysis for mesenchymal stem and neuronal markers was performed on neuro-induced DPSCs with EGF and bFGF.

### 2.8. Reverse Transcription-Quantitative PCR (RT-qPCR) Analysis

DPSCs in hypoxic condition (Table 1) were analyzed by RT-qPCR analysis. Total cellular RNA was extracted from 100,000 DPSCs using TRIzol^®^ Reagent (Thermo Fisher Scientific, Rockford, IL, USA), and its quality and quantity were evaluated on a NanoDrop spectrophotometer (Thermo Fisher Scientific, Rockford, IL, USA). For the qPCR assay, cDNA was synthesized from 500 ng of total RNA with SuperScript (Euroclone, Milan, Italy). Next, the qPCR was carried out using SYBR green master mix (Luna), according to the manufacturer’s protocols. The amplification reaction was performed on a MiniOpticon Real-Time PCR System (Bio-Rad, Milan, Italy) using the following program: the RT reaction was set at an initial denaturation step at 95 °C for 1 min, followed by 95 °C for 15 s. The reaction mixture was heated according to the Tm shown in Table 3 for 30 s, followed by amplification that consisted of 40 cycles of denaturation at 95 °C for 15 s, and annealing and extension at 60 °C for 30 s, according to the manufacturer’s protocol. The relative expressions of the neuronal marker genes investigated were calculated using the relative quantification 2^(−∆∆C(T))^ 2^−∆∆Cq^ method [58], with *GAPDH* as a reference gene commonly used for this purpose. The primer sequences used in this RT-qPCR analysis are listed in Table 3.

### 2.9. ELISA for EGF and bFGF

We evaluated, by ELISA, EGF and bFGF concentration in the CM of DPSCs kept for 5 and 16 days in hypoxia (Table 1). The quantity of EGF and bFGF, in each sample, was evaluated using the Human EGF ELISA Kit and the Human FGF basic ELISA Kit (Abcam, Cambridge, MA, USA), following the manufacturer’s instructions. All experimental points were analyzed in triplicate.

### 2.10. Immunofluorescence Analysis

SH-SY5Y and DPSCs were seeded at a density of 2 × 10^4^ cells/mL in 6-well plates and standard culture medium for 24 h. After seeding, both cell lines were stimulated for 48 h and 10 days with DPSC-CM, as reported in Table 2 and incubated in conditions of hypoxia and normoxia for 5 and 16 days. At the end of the treatment, the obtained samples were used for immunofluorescence analysis. Briefly, SH-SY5Y and DPSCs treated as above were fixed with 4% paraformaldehyde for 10 min at 4 °C and permeabilized with permeabilizing solution (0.1% Triton X-100 in PBS) for 10 min at 4 °C. After washing in PBS, cells were incubated with mouse Anti-β3-tubulin mAb and mouse Anti-NFH mAb (Cell Signaling Technology, Danvers, MA, USA) overnight at 4 °C, followed by Anti-mouse IgG Alexa fluor 488 (ThermoFisher Scientific, Rockford, IL, USA) in the dark for an additional 60 min. Finally, cells were observed with a Zeiss Axio Vert. A1 fluorescence microscope (Zeiss, Milan, Italy). Quantitative analysis of the fluorescence intensity for each protein of interest was measured by Image J.

### 2.11. Statistical Analysis

The data are expressed as mean ± standard error (SE) obtained from 3 experimental replicates (experimental replicates = 3 for each treatment/time point). Statistical analysis was performed using Prism 6 (GraphPad). Student’s *t*-test or one-way ANOVA were used for multiple comparisons and performed on data sets. Values were considered statistically significant at *p* ≤ 0.05.

## 3. Results

### 3.1. Determination of Hypoxia Effects on the Proliferation and Morphological Features of DPSCs

We studied the morphology and the growth potential of DPSCs exposed or not to hypoxia for 5 and 16 days, as previously described. As reported in Figure 1, representative images of DPSCs exposed to hypoxia for 5 days (5H) showed a slight change in morphology compared to normoxic DPSCs (5N) (Figure 1A). 5H appeared morphologically quite like 5N, even though hypoxia seemed to induce an increase in actin expression compared with normoxic cells. Moreover, at this time point, hypoxia did not seem to significantly influence DPSC growth (Figure 1B). DPSCs exposed to hypoxia for 16 days (16H) were thin and stretched (Figure 1A). Furthermore, the 16H proliferation rate was significantly decreased with respect to the other culture conditions (Figure 1B).

### 3.2. Comparative Characterization of DPSCs’ Stem and Neuronal Markers by Flow Cytometry

To evaluate a possible role of hypoxic conditioning in the induction of neuronal differentiation, DPSCs were exposed for 5 and 16 days to hypoxia at 1% O_2_ (5H, 16H) or maintained in normoxia at 21% O_2_ (5N, 16N). Figure 2A shows that all mesenchymal stem cell markers (CD44, CD90, CD105, STRO1, and CD73) commonly expressed by DPSCs [59] decreased in the cells exposed to hypoxia after 5 days. These markers decreased from 69.5, 70.8, 34.2, 68.1 and 74.4% to 76.4, 75.1, 63.9, 73.4 and 84.22% in the normoxic cells, respectively. In particular, CD105 showed the greatest decrease (47%). The evaluation of the same stem cell markers at 16 days (16H and 16N) showed an even larger decrease. Interestingly stem cells markers in 16H decreased greatly, reaching values close to zero: 2.4, 2, 0, 0.1 and 15.5% compared to 73.6, 72.1, 63.7, 61.3 and 75.8% for 16N. Conversely, we noted a slight but significant increase in neuronal markers from 44.7, 1.8, 2.2, and 3.7% of 5N to 56.6, 9.1, 5.00, and 17.1% of 5H. A more relevant increase in these markers was observed after 16 days, shifting from values of 2.6, 2.9 and 9.3% (16N) to values of 53.3, 47.1 and 43.3% (16H). Only the nestin marker decreased from 16N to 16H, suggesting the specific role of this marker in neuronal differentiation. Moreover, the phenotypic expression of 16H was very similar to DPSCs treated with EGF and bFGF factors for 14 days, a culture condition that determines the neuronal differentiation of DPSCs, as shown Figure 2B [13].

### 3.3. The Profiling of Differential mRNA Expressed in DPSCs under Hypoxic or Normoxic Conditions

To confirm the role of hypoxia in determining a phenotypic change in DPSCs, we compared the mRNA expression profile of the DPSCs exposed (5H, 16H), or not (5N, 16N), to hypoxia. The levels of expression of the early (nestin, GFAP, bFGF and EFG) and late markers nerve growth factor (NGF), brain-derived neurotrofic factor (BDNF) and glial cell line-derived neurotrofic factor (GDNF) of neural differentiation were assessed. Real-time *RT-qPCR* showed that most of the marker genes were upregulated after 5 days (Figure 3A). Only bFGF did not seem upregulated at this time point. Interestingly, the marked increase in nestin and GFAP, as well as the GDNF and BDNF mRNA expression observed in 5H, suggests that these cells may be derived from neural crest origin, and is thus easily addressable vs. a neurogenic phenotype.

As shown, hypoxia at 16 days determined a new profound change in mRNA marker expression levels. The levels of expression of nestin, EGF, BDNF, GFAP and GDNF were significantly downregulated in 16H cells compared with 5H cells (Figure 3A). Conversely, NGF and bFGF mRNA levels were strongly increased compared with 5H cells, reaching values of 4-fold and 16-fold greater than the controls, respectively.

### 3.4. Growth Factor Expression in Hypoxic or Normoxic DPSCs’ Conditioned Media (CM)

Based on our results, hypoxia determined a switch in the expression profiles of DPSCs, addressing them vs. the neurogenic phenotype. In particular, the levels of mRNA expression of EGF and bFGF at 5 days and 16 days of hypoxia, respectively, were particularly increased. Therefore, we investigated if a correspondent increase in these secreted growth factors was detectable in the conditioned media of DPSCs exposed or not exposed to hypoxia for 5 days and 16 days by human EGF and bFGF basic ELISA Kits (Abcam), according to the manufacturer’s instructions.

The analysis of the CM from the DPSCs stimulated with hypoxia for 5 days showed a significant increase in EGF expression, confirming the data obtained on mRNA profile expression. EGF increased even more in CM from DPSCs stimulated with hypoxia for 16 days, probably because of the observed increase in mRNA at 5 days of hypoxia (Figure 3B). bFGF secretion was highly detectable in the CM of DPSCs treated with hypoxia for 16 days in parallel with its mRNA expression.

### 3.5. Analysis of the Paracrine Induction of the Neuronal Phenotype by Immunocytochemistry and Flow Cytometry

To verify the ability of DPSCs to induce neuronal differentiation by an autocrine/paracrine mechanism mediated by growth factor secretion, we verified the ability of DPSC-CM (5H, 16H) to differentiate SH-SY5Y neuroblastoma cells into cells possessing a more mature, neuron-like phenotype. As demonstrated by several authors, SH-SY5Y cells, in their undifferentiated form, express immature neuronal markers and lack mature neuronal markers, but following treatment with differentiation-inducing agents, SH-SY5Y cells become morphologically more like primary neurons [60].

When exposed to 5H and 16H CM, SH-SY5Y showed different phenotypical changes compared to 5N- and 16N-CM-exposed cells. As shown by immunofluorescence, the neuronal markers NFH and β3-Tubulin were highly expressed in all groups of SH-SY5Y treated with DPSCs’ hypoxic CM (5H, 16H) in comparison with groups of cells treated with DPSCs’ normoxic CM (5N, 16N) (Figure 4A). Moreover, the analysis by flow cytometry of NFH and β3-Tubulin expression in SH-SY5Y-treated or untreated cells confirmed, in a quantitative way, the results obtained in immunofluorescence. In SH-SY5Y exposed for 10 days with 16H CM, the neuronal marker expression of NFH and β3-Tubulin reached values of 96.1% and 92.8%, respectively (Figure 4B).

Similarly, we tested the ability of 5H and 16H CM DPSCs on the differentiation of undifferentiated DPSCs vs. a neuronal phenotype. As highlighted in the immunofluorescence images, the neuronal markers NFH and β3-Tubulin were markedly increased in DPSCs treated with hypoxic DPSC-CM (5H CM and 16H CM), compared with normoxic DPSC-CM (5N CM and 16N CM). In particular, the increase in neuronal marker expression was visible in the DPSCs’ long processes, such as those of primary neurons (Figure 5).

### 3.6. Phenotype Characterization of DPSCs by Flow Cytometry

The DPSCs’ treated groups, 5H CM and 16H CM, were compared with the 5H and 16H groups by flow cytometry to determine the profile of expression of several neuronal and stem markers. As shown in Figure 6, the stem cells markers CD44 and CD73 were still quite high in the 5H group. Only CD90 was lowered in D10d-5H. Moreover, the neuronal markers investigated (nestin, β3-tubulin, NFH and GAP43) were still low and similarly expressed in the 5H and in 5H CM groups.

Intriguingly, the 16H CM groups showed a dramatic decrease in stem cells markers: the 16H group reached values below 10% positivity. Conversely, neural stem cells markers strongly increased in 16H CM, reaching values higher than 16H.

## 4. Discussion

In the present work, we have shown in vitro that hypoxia conditioning can induce a commitment of DPSCs vs. a neuronal phenotype. Intriguingly, the hypoxia-mediated differentiation of DPSCs was probably induced through an autocrine/paracrine mechanism.

We first demonstrated that hypoxia could induce a phenotypic differentiation of DPSCs by evaluating morphology and growth potential and characterizing the DPSCs’ stem and neuronal markers by flow cytometry in comparison to the mRNA expression profiles of DPSCs exposed or not exposed to hypoxia.

Then, we investigated the autocrine/paracrine effect of hypoxia on the DPSCs’ neuronal commitment by the analysis of immunocytochemistry and flow cytometry of the stem and neuronal markers. Moreover, we assessed the mRNA profiling of DPSCs treated with CM enriched by hypoxia.

DPSCs, probably likely because of their ectodermal origin, are easily addressable into neuron-like cells and, after neuronal differentiation, these cells even exhibit typical electrophysiological properties [34,36,38]. Several studies indicate that intrinsic positional information can influence cell phenotype, and that neural crest populations can also produce mesenchymal derivatives confirming the role of the environment in orchestrating signals for neural crest development and differentiation [15,61]. Moreover, it is known that DPSCs express nestin, and low basal levels of markers associated with mature CNS cell types, such as the neuronal markers microtubule-associated protein 2 (MAP2), NF, β3-Tubulin, CNPase associated with oligodendrocytes, and astrocytic marker GFAP [35,36]. The data reported in the literature confirm that DPSC populations can be differentiated into neuron-like cells under appropriate conditions [9,14]. Of these conditions, O_2_ concentration is one of the most important critical factors that could play a fundamental role during DPSC differentiation. It has been demonstrated that O_2_ concentration in the culture environment is essential to maintain stem cell plasticity and proliferation [44]. Moreover, it is also known that oxygen affects cellular characteristics and tissue-remodeling processes, having the potential role to decide cell fate [28,44]. In recent studies, researchers have demonstrated the effects of hypoxia on DPSCs’ stemness, in addition to inducing the proliferation of MSCs under certain conditions [47,48,62].

Other researchers have demonstrated an opposite effect of hypoxia at 1% O_2_ concentration showing a reduced proliferation rate of MSCs with arrest in the G0/G1 phase of the cell cycle [63]. Moreover, hypoxic conditions can promote or inhibit DPSC proliferation and/or survival through various cellular responses, probably depending on conditions such as a reduction in oxygen tension.

A recent report by Zainal Ariffin et al. demonstrated that, under neurogenic stimuli induction, differentiated DPSCs showed a reduction in growth rate after 24 h, whereas the undifferentiated cells maintained their growth rate [64].

According to these literature data, our results show that DPSCs, when exposed to hypoxia, change their morphology: growth potential decreased after 16 days of treatment at 1% O_2_. Moreover, when exposed to hypoxia 1%, DPSCs, at both times considered, were induced to differentiate vs. a neuronal phenotype, showing a switch, strongly relevant after 16 days, of mesenchymal stem cells markers CD44, CD90, CD105, and CD73, commonly expressed by DPSCs vs. neuronal markers nestin, β3-Tubulin, NFH, and GAP 43 [59]. The commitment vs. a neuronal phenotype was also confirmed by the mRNA expression profile of the DPSCs exposed (5H, 16H), or not (5N, 16N), to hypoxia. The levels of expression of the early (nestin, GFAP, bFGF, and EFG) and late markers (NGF, BDNF, and GDNF) of neural differentiation showed that most of these marker genes were upregulated after 5 days. bFGF was upregulated only after 16 days of exposure. Interestingly, the marked increase in nestin and GFAP, as well as GDNF and BDNF mRNA expression observed in DPSCs treated for 5 days, suggested that these cells may be easily addressable vs. a neurogenic phenotype, probably because of their neural crest origin, as also suggested in several recent works [37]. The upregulation in EGF and bFGF mRNA levels was confirmed by the analysis of these secreted growth factors in the conditioned media of DPSCs exposed, or not, to hypoxia. In fact, 5H-DPSCs CM showed a significant increase in EGF expression. EGF increased even more in 16H-DPSCs’ CM, probably because of the observed increase in mRNA at 5 days of hypoxia. bFGF secretion was highly detectable in 16H-DPSCs CM, in parallel with its mRNA expression.

Our data are in accordance with several studies which demonstrate the possibility to differentiate DPSCs vs. a neuronal phenotype by paracrine induction mediated by growth factors. For example, Vaseenon et al. reported that bFGF can enhance peripheral nerve regeneration after injury [39,40]. Similarly to our results, Rafiee et al. demonstrated that both EGF and bFGF can induce DPSC neuronal differentiation [41]. Zhang et al. revealed that bFGF and NGF increased DPSC neural differentiation synergistically [65]. Moreover, other authors have demonstrated that the treatment of DPSCs with bFGF led to their proliferation and differentiation during neurogenesis [66].

We first verified the hypothesis of a paracrine mechanism mediated by growth factors using CM obtained from hypoxia pretreated DPSCs to differentiate SH-SY5Y neuroblastoma cells. As demonstrated by several authors, SH-SY5Y cells become more like primary neurons following treatment with differentiation-inducing agents. In this way, they shift from the undifferentiated form towards a more mature, neuron-like phenotype, expressing neuronal markers and the loss of markers of immature neurons [62,67].

Immunofluorescence and mRNA profile analysis revealed a marked expression of the neuronal markers NFH and β3-Tubulin in SH-SY5Y exposed for 10 days with 16H CM, reaching values of 96.1% and 92.8%, respectively.

Similarly, DPSCs treated with hypoxic CM (16H CM) also showed a profound modification of mRNA expression profile, shifting from a stem cells phenotype towards a neural phenotype. Intriguingly, DPSCs treated with hypoxic CM reached expression levels of neuronal markers higher than DPSCs treated directly with hypoxia for 16 days.

Our results match with recent studies, demonstrating the neuroprotective function of DPSCs related to a high expression of trophic factors, including BDNF, GDNF, NGF, and NT-3 [36,68]. These factors seem to promote the growth of neurons, induce neurogenesis at the site of injury, and play a pivotal role in neuroprotection against neurodegeneration and neuron function recovery [69].

These abilities make DPSCs an efficient cellular model for the study of neurodegenerative diseases and could contribute to autologous and allogenic cell therapy for CNS injury [36,60,69,70,71,72].

Moreover, according to recent studies showing the therapeutic activity of MSCs due to the release of their paracrine factors in the CM, CM from differentiated neuronal DPSCs could be inoculated in vivo with therapeutic effects in neurodegenerative disease models [51,68].

Similarly, an interesting paper provided evidence that DPSCs, through the secretion of neurotrophins, significantly increased both the survival and neurito-genesis of primary adult rat β3-tubulin retinal cells in an in vitro co-culture assay. Furthermore, when transplanted into the vitreous body of adult rats after optic nerve crush (ONC), DPSCs significantly promoted Brn-3aþRGC survival and axon regeneration [73]. Thus, DPSCs, thanks to their ability to differentiate into various cell types, including cells expressing RPE, photoreceptor cell markers, and the cells of the retina, could also be used for the regeneration of diseased retina.

Although some studies show a direct therapeutic effect of MSCs dependent on their abilities of “engraftment” and differentiation, other studies have shown that, probably due to reduced survival and implantation capacity, the actual impact of these cells in vivo is scarce, limiting their potential for cell replacement therapy [74,75,76]. The risks to research participants undergoing stem cell transplantation include side effects such as tumor formation, inappropriate stem cell migration, or immune rejection of the transplanted stem cells.

In contrast, stem cell’s use to provide therapeutic factors able to prevent disease progression is becoming a more realistic and achievable goal. Thus, as has already been demonstrated in several works, DPSCs may act mainly in a paracrine way through the inoculation of their secretome. For these reasons, we decided to test the conditioned medium of DPSCs to evaluate whether the factors released by these cells subjected to different stimulation conditions could have a neuroprotective and regenerative role in neurodegenerative disease models.

Further investigations and future clinical trials to provide essential information on the best method for producing DPSC secretomes through hypoxic stimulation (time and O_2_ concentration and/or serum deprivation) and their efficacy are needed. Moreover, the object of our next work will be to focus on the signaling pathways of DPSCs activated by hypoxia, and on the roles of bFGF and EGF signaling in the differentiation of DPSCs.

## 5. Conclusions

In conclusion, we demonstrated that hypoxia (O_2_ 1%), under specific in vitro conditions, may exert on DPSCs a commitment towards a neurogenic phenotype, and it may stimulate the secretion of multiple growth factors that are, at least in part, responsible for promoting the neuronal differentiation of undifferentiated and/or partially committed cells. Our results highlight the potential of DPSCs as promising candidates for stem cell therapy in future therapeutical approaches for neurological diseases. Since we have demonstrated that DPSCs stimulated by hypoxic CM exhibited a neuronal expression profile of neural stem cell markers higher than DPSCs treated with normoxic CM, these results suggest the potential clinical utility not only of differentiated neuronal DPSCs, but also of CM-derived DPSCs which can be inoculated in vivo with therapeutic effects in neurodegenerative disease models. Moreover, the implementation of such a strategy may be beneficial because, being a cell-free product, it should be more easily managed as a medicine.

## Figures and Tables

**Figure 1 biomedicines-10-01056-f001:**
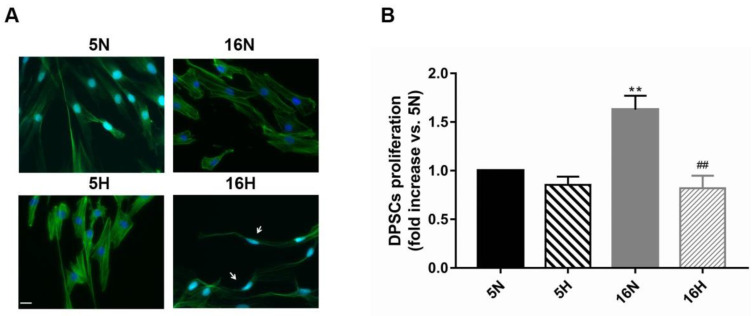
Effects of hypoxia on DPSC morphology and proliferation. (**A**) Representative images of phalloidin-stained DPSCs exposed or not to hypoxia for 5 and 16 days, as described in the Materials and Methods section (Table 1). White arrows indicate cell morphology changes. Scale bar 20 µm. (**B**) Histograms show that hypoxia induced a significant decrease in proliferation rate exclusively after 16 days of exposure in comparison to normoxia. Histograms indicate the means ± SE of three different cultures, each of which was tested in triplicate. Statistical analysis was performed by one-way ANOVA followed by Tukey’s test. ∗∗ *p* < 0.01 vs. 5N, and ## *p* < 0.01 vs. 16N.

**Figure 2 biomedicines-10-01056-f002:**
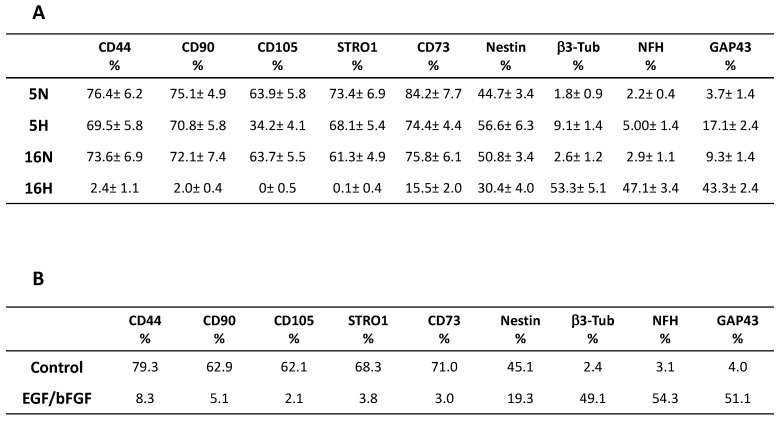
Effects of hypoxia on phenotypic expression profile of DPSCs. (**A**) shows that mesenchymal stem cells markers CD44, CD90, CD105, STRO1 and CD73 decreased in cells exposed to hypoxia at 5 and 16 days of exposure. Conversely, neuronal markers increased after hypoxia exposure. Phenotypic expression of 16H was very similar to (**B**) DPSCs treated with EGF and bFGF factors for 14 days, a culture condition that determines the neuronal differentiation of DPSCs.

**Figure 3 biomedicines-10-01056-f003:**
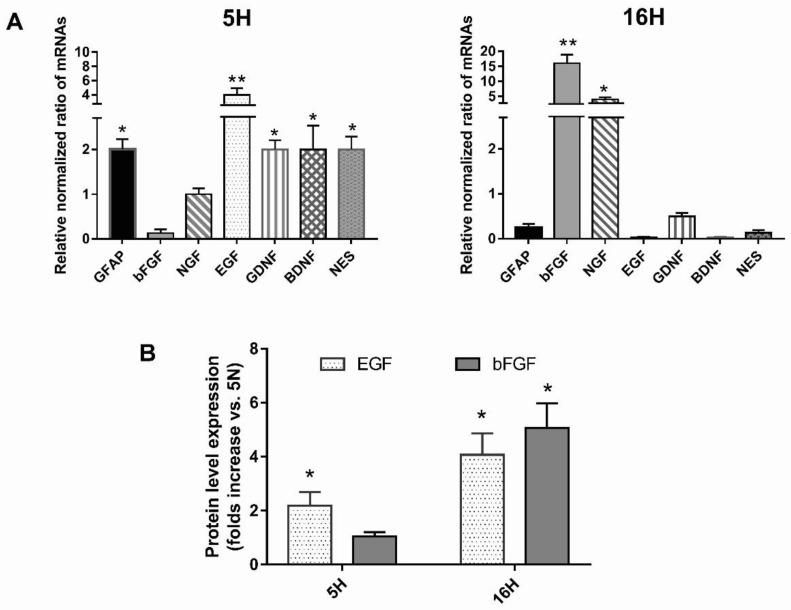
Effects of hypoxia on the mRNA profile expression of DPSCs. (**A**) Real-time RT-qPCR analysis of the levels of expression of early and late markers of neural differentiation (nestin, GFAP, EGF, NGF, bFGF BDNF, GDNF) after 5 days and after 16 days of hypoxic exposure. (**B**) Secreted growth factors bFGF and EGF were detected in the conditioned media of DPSCs exposed or not exposed to hypoxia for 5 days and 16 days by human EGF and FGF basic ELISA Kits. Histograms indicate the means ± SE of three different cultures, each of which was tested in triplicate. Statistical analysis was performed by Student’s *t*-test; * *p* < 0.05; ** *p* < 0.01 vs. 5N.

**Figure 4 biomedicines-10-01056-f004:**
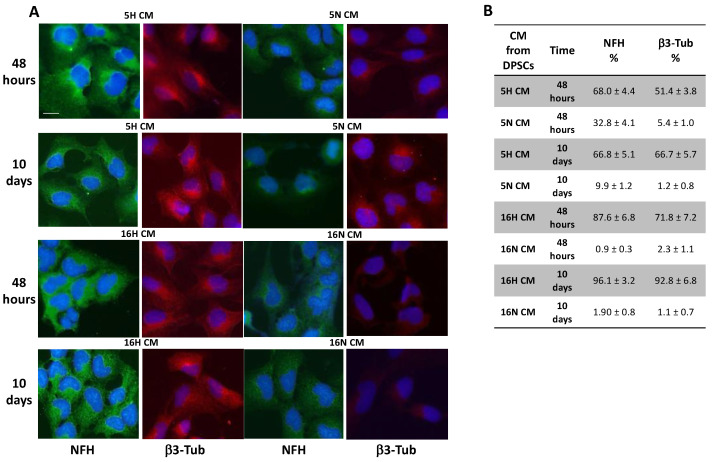
Immunofluorescence and flow cytometry analysis of SH-SY5Y. (**A**) SH-SY5Y exposed to DPSC-CM (5H, 16H) were stained for neuronal markers NFH and β3-Tubulin that were highly expressed in all groups of SH-SY5Y treated with DPSCs’ hypoxic CM (5H, 16H) in comparison with groups of cells treated with DPSCs’ normoxic CM (5N, 16N). (**B**) Flow cytometry analysis of NFH and β3-Tubulin expression in SH-SY5Y-treated groups or untreated cells. Scale bar 20 µm.

**Figure 5 biomedicines-10-01056-f005:**
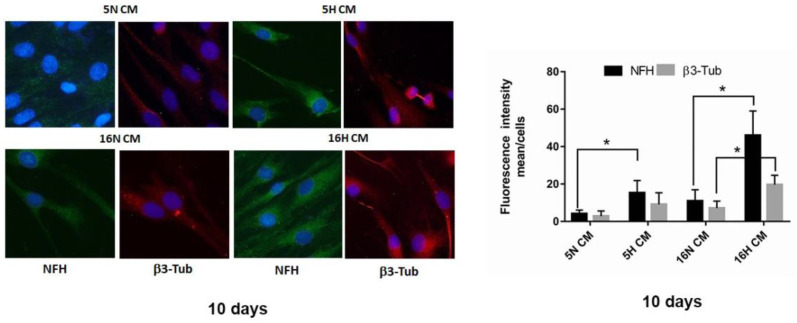
Immunofluorescence analysis of DPSCs. (**left**). Immunofluorescence analysis of NFH and β3-Tubulin in DPSCs treated with DPSC-CM (5H CM, 16H CM) or DPSC-CM (5N CM, 16N CM) groups. Scale bar 20 µm. (**right**). Analysis of fluorescence intensity NFH and β3-Tubulin in DPSCs treated with DPSC-CM (5H CM, 16H CM) or DPSC-CM (5N CM, 16N CM) groups. Data are represented as mean ± SE. Statistical analysis was performed by Student’s *t*-test; * *p* < 0.05; 5H and 16H vs. 5N and 16N.

**Figure 6 biomedicines-10-01056-f006:**
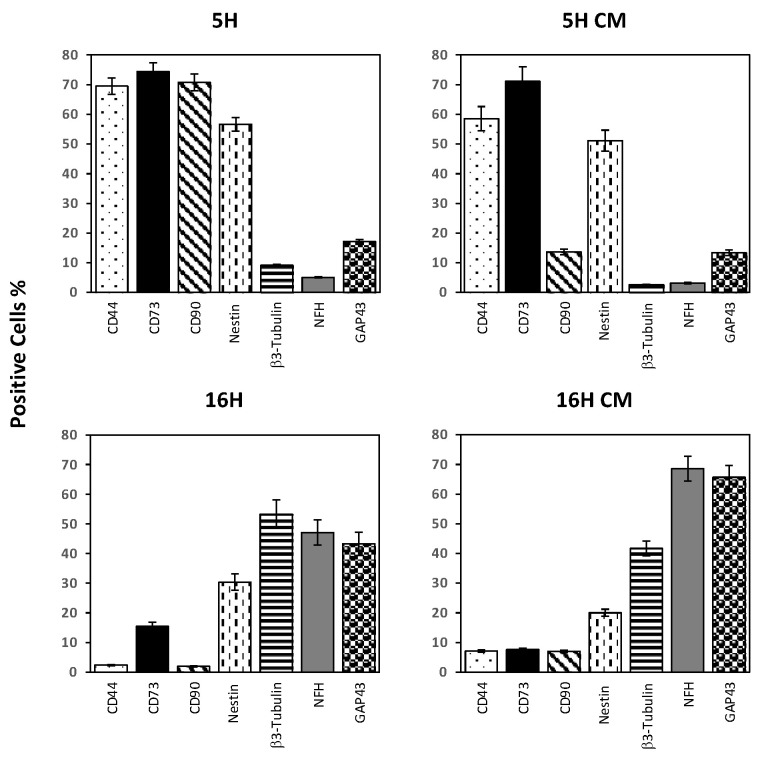
Effects of hypoxia on the phenotypic expression profiles of DPSCs treated with CM enriched by hypoxia. Flow cytometry analysis of DPSC-treated groups; 5H CM and 16H CM were compared with the 5H and 16H groups to determine the profile of expression of several neuronal and stem markers. Histograms indicate the means ± SE of three different experiments.

**Table 1 biomedicines-10-01056-t001:** Summary of culture conditions.

Abbreviations	Type of Treatment
5N	DPSCs for 5 days in normoxia
5H	DPSCs for 5 days in hypoxia
16N	DPSCs for 16 days in normoxia
16H	DPSCs for 16 days in hypoxia

**Table 2 biomedicines-10-01056-t002:** Summary of conditioned media treatment.

Abbreviations	Type of Conditioned Media
5N CM	Conditioned media derived from normoxic DPSCs for 5 days
5H CM	Conditioned media derived from hypoxic DPSCs for 5 days
16N CM	Conditioned media derived from normoxic DPSCs for 16 days
16H CM	Conditioned media derived from hypoxic DPSCs for 16 days

**Table 3 biomedicines-10-01056-t003:** Primer sequences used in RT-qPCR analysis.

Gene	Forward Primer	Reverse Primer	Tm Value
*GAPDH*	AGGTGAAGGTCGGAGTCAAC	CCATGTAGTTGAGGTCAATGAAG	58–65
*GFAP*	TAGAGGGCGAGGAGAACCG	GTGGCCTTCTGACACAGACTTG	64
*bFGF*	CTGTACTGCAAAAACGGG	AAAGTATAGCTTTCTGCC	63
*NGF*	CATGCTGGACCCAAGCTCA	GACATTACGCTATGCACCTCAGTG	60
*EGF*	GGTCAATGCAACCAACTTCA	GGCATTGAGTAGGTGATTAG	63
*GDNF*	TCAAATATGCCAGAGGATTATCCTG	GCCATTTGTTTATCTGGTGACCTT	64
*BDNF*	AGCTCCGGGTTGGTATACT	CCTGGTGGAACTTCTTTGCG	64
*NES*	TGGCAAAGGAGCCTACTCCAAGAA	ATCGGGATTCAGCTGACTTAGCCT	65

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
