# Peer review of "Hypoxia Induces DPSC Differentiation versus a Neurogenic Phenotype by the Paracrine Mechanism"

_biomedicines, 2022, doi:10.3390/biomedicines10051056_

Round 1

Reviewer 1 Report

In the current  manuscript the authors suggested that hypoxia seems favoring DPSCs differentiation into neuron-like cells under  specific conditions, suggesting the potential clinical utility both differentiated neuronal DPSCs and CM derived hypoxic DPSCs.

Overall the paper is well structured, the conclusions are supported by the results obtained, but still a paper need a major revisions. 

major concerns:

the abstract need to be revised.

it is strongly suggest to avoid to start an abstract with " thanks to" doesn't have a scientific soundness.  

the aim need to be included in the abstract and the abstract need a substantial revision. 

It is strongly suggest to read the following paper entitled "Ascorbic Acid: A New Player of Epigenetic Regulation in LPS gingivalis Treated Human Periodontal Ligament Stem Cells" , "Oral bone tissue regeneration: Mesenchymal stem cells, secretome, and biomaterials" to add some references regarding MSCs properties " and " Decellularized Dental Pulp, Extracellular vesicles  and 5-Azacytide: a new tool for endodotic regeneration"

the immunofluorescence images need to be improved, they look out of focus and the scale bar is missing, please revised. 

the paper need to revised by a native english speaker

Author Response

General comments to reviewers:

Overall, we revised all the contents based on comments from reviewers. We hope this version of the manuscript is satisfactory to reviewers and editors. We highlighted with yellow color revised parts.

Reviewer 1

In the current  manuscript the authors suggested that hypoxia seems favoring DPSCs differentiation into neuron-like cells under  specific conditions, suggesting the potential clinical utility both differentiated neuronal DPSCs and CM derived hypoxic DPSCs.

Overall the paper is well structured, the conclusions are supported by the results obtained, but still a paper need a major revisions.

major concerns:

the abstract need to be revised.

it is strongly suggest to avoid to start an abstract with " thanks to" doesn't have a scientific soundness. 

the aim need to be included in the abstract and the abstract need a substantial revision.

As correctly suggested by the referee, we made a substantial revision of the abstract. We clarified the aim, and we revised the English language.

            -see Abstract page 1.

It is strongly suggest to read the following paper entitled "Ascorbic Acid: A New Player of Epigenetic Regulation in LPS gingivalis Treated Human Periodontal Ligament Stem Cells" , "Oral bone tissue regeneration: Mesenchymal stem cells, secretome, and biomaterials" to add some references regarding MSCs properties " and " Decellularized Dental Pulp, Extracellular vesicles  and 5-Azacytide: a new tool for endodotic regeneration"

               We double-checked and updated the reference section with recent and relevant articles as suggested by the reviewer.

                              -see manuscript and References session.

the immunofluorescence images need to be improved, they look out of focus and the scale bar is missing, please revised.

               - Likely, images display low quality since they have been added in the text.  However, we will send the high resolution images in a separate file in the resubmission process.

the paper need to revised by a native english speaker

- According to the reviewer's recommendation, we performed a revision of the English by a native English speaker.

Reviewer 2 Report

"Hypoxia induces DPSCs differentiation versus a neurogenic phenotype by paracrine mechanism"

It is very interesting to shown in vitro that hypoxia conditioning is able to induce a commitment of DPSCs vs a neuronal phenotype. However, there are a few corrections that are essential to meet the standard for publication. Please refer to the following comments.

Please add the statistical evaluation method in the method column.

Table 2. Summary of conditioned media treatment Please make it easier to see.

It's very complicated and confusing.

It may be clinically applicable to your research. Please add to the discussion section about the specific future prospects of this research.

Many of the references in your dissertation are old. Please update to new bibliographic citations whenever possible.

Author Response

General comments to reviewers:

Overall, we revised all the contents based on comments from reviewers. We hope this version of the manuscript is satisfactory to reviewers and editors. We highlighted with yellow color revised parts.

Reviewer 2

"Hypoxia induces DPSCs differentiation versus a neurogenic phenotype by paracrine mechanism"

It is very interesting to shown in vitro that hypoxia conditioning is able to induce a commitment of DPSCs vs a neuronal phenotype. However, there are a few corrections that are essential to meet the standard for publication. Please refer to the following comments.

Please add the statistical evaluation method in the method column.

               Thank you for pointing out our omission. As correctly suggested we added the paragraph of Statistical Analysis.

               See materials and methods session page 7

Table 2. Summary of conditioned media treatment Please make it easier to see.  

It's very complicated and confusing.

Thank you for your correct suggestion. We realized that the table is not self-explanatory and easily understood. So, we modified and simplified it, revising figures and results accordingly.

See table 1 and table 2 in materials and methods session page 5

It may be clinically applicable to your research. Please add to the discussion section about the specific future prospects of this research.

               We revised the discussion session to highlight future prospects of our research

See discussion session page 15

Many of the references in your dissertation are old. Please update to new bibliographic citations whenever possible.

               As suggested, we updated the references in the text, adding new bibliographic citations

               See references session page 17

Reviewer 3 Report

The authors in the manuscript show that hypoxia induces differentiation of DPSC but fail to elucidate the underlying mechanism or the molecules involved. The data shown is also very preliminary and as such the authors have to identify the factors responsible for promoting such differentiation.

Some of the figures can be combined and as such do not warrant individual panels. The images provide do not convincingly demonstrate the conclusions drawn. 

Fig 1A, the authors vaguely say there are morphology al changes without characterizing them. IT will be good to stain the cells for actin and nucleus and characterize the morphology in a better way. Fig 2 do not have any statistical tests done to provide levels of significance. Performing western blots is a better way to show the data. Figure 2 and 3 can be combined into 1 figure. Similarly, fig 4 and 5 can be combined into a single figure. Figure 5 needs to be quantified with statistical analyses.

How does hypoxia lead to differentiation?  Is stabilization of Hif1 required for the process. The authors need to clearly demonstrate:

a: The role of hypoxia in the process

b: The specific factor(s) required for the autocrine/paracrine function.

Author Response

General comments to reviewers:

Overall, we revised all the contents based on comments from reviewers. We hope this version of the manuscript is satisfactory to reviewers and editors. We highlighted with yellow color revised parts.

Reviewer 3.

 The authors in the manuscript show that hypoxia induces differentiation of DPSC but fail to elucidate the underlying mechanism or the molecules involved. The data shown is also very preliminary and as such the authors have to identify the factors responsible for promoting such differentiation.

               Actually, we realize that in the present work we have not provided clarifications on the action mechanism responsible for the differentiation but  the aim of our study is to demonstrate that conditioned media derived from DPSCs stimulated with hypoxia are able to stimulate the differentiation of undifferentiated cells. We demonstrated the over-production of some factors. Among these, the EGF and the FGF seem to be the most involved. In any case, in a future work we will investigate the pathways activated by these factors in undifferentiated DPSCs.

Some of the figures can be combined and as such do not warrant individual panels. The images provide do not convincingly demonstrate the conclusions drawn.

Since the combining multiple images could worsen the strenght and visibility of the data, we prefer to keep them separate.

Fig 1A, the authors vaguely say there are morphology al changes without characterizing them. IT will be good to stain the cells for actin and nucleus and characterize the morphology in a better way. Fig 2 do not have any statistical tests done to provide levels of significance. Performing western blots is a better way to show the data. Figure 2 and 3 can be combined into 1 figure. Similarly, fig 4 and 5 can be combined into a single figure. Figure 5 needs to be quantified with statistical analyses.

               According to the reviewer’s suggestion, cells were stained with Phalloidin to visualize cellular cytoskeleton, nuclei were counterstained with Hoechst and images were acquired.

               See materials methods and results section;

                                             -page 4

-page 8 (Figure 1 panel A)

How does hypoxia lead to differentiation?  Is stabilization of Hif1 required for the process. The authors need to clearly demonstrate:

a: The role of hypoxia in the process

b: The specific factor(s) required for the autocrine/paracrine function.

            As suggested by the reviewer, the stabilization of hif1α could be required as key molecule involved in the differentiation of DPSCs. As reported by several authors, HIF1alfa is expressed also in normoxic MSCs, and it has a crucial role during differentiation of these cells. The MSCs are able to maintain an undifferentiated state and glycolytic energy production in normoxia conditions and under variable oxygen conditions by means of high mRNA expression and the stabilization of HIF -1α. During differentiation instead, MSCs require the sufficient blood and oxygen concentration. So their differentiation can be accompanied  by reduced HIF stabilization and a metabolic shift to oxidative phosphorylation in the MSCs (10.3390/cells10082161, 10.1002/stem.1435.). According to above cited references we observed that hypoxia determined an increase of Hif1 alfa expression correspondent to HIF protein stabilization until 5 days of exposure (5H). This increase was demonstrated in mRNA expression and confirmed also by western blotting assay. With prolonged hypoxia exposure (16 days) we obtained a down regulation of mRNA and protein expression of HIF1α probably because at this time DPSCs resulted more committed versus a neuronal-like phenotype.

                        See Figure A

In conclusion, based on these data, we could only postulate a mechanism in which HIF1α decrease depends from metabolic shift. In a future study, we will investigate the link between HIF and DPSC metabolism under prolonged hypoxia exposure.

Round 2

Reviewer 1 Report

The authors replied point by point to all comments done.

Author Response

We thanks the reviewer for its answer

Reviewer 2 Report

Thank you for giving me this opportunity to re-review your revised manuscript.

I am happy that all of the suggested corrections have been made.

Thank you for spending so much time for revised manuscript.

Author Response

We thanks the reviewer for its answer, we are glad to have satisfied all your requests

Reviewer 3 Report

The authors fail to provide experimental evidence for most of my concerns; I do not deem the manuscript to be accepted in the present form.

Author Response

We are surprised about your response because we believe we have made most of the requested changes and responded to comments.

Moreover, with respect to the previous version and according to Reviewer’s request we also added the fluorescence intensity analysis in Figure 5.